# From hospital to home: A heightened window of vulnerability post-critical illness

**Amanda Fuchsia Howard**[1]*, **Kelsey Lynch**[1], **Sally Thorne**[1], **Leanne M. Currie**[1], **Rakesh C. Arora**[2,3], **Robert C. McDermid**[4,5], **Omar Ahmad**[6], **Sarah Crowe**[1,5], **Sybil Hoiss**[7], **Anita David**[1], **Joanne Chu**[1], **Alice Erchov**[1], **Brianna Hou**[1], **Arnima Singh**[1], **Miki Tsui**[1], **Gregory Haljan**[4,5]

**1** School of Nursing, The University of British Columbia, Vancouver, British Columbia, Canada, **2** Harrington Heart and Vascular Institute, University Hospitals, Cleveland, Ohio, United States of America, **3** Department of Surgery, Division of Cardiac Surgery, Case Western Reserve University, Cleveland, Ohio, United States of America, **4** Faculty of Medicine, The University of British Columbia, Vancouver, British Columbia, Canada, **5** Fraser Health, Surrey, British Columbia, Canada, **6** Island Health, Victoria, British Columbia, Canada, **7** Provincial Health Services Authority, Vancouver, British Columbia, Canada

* fuchsia.howard@ubc.ca (AFH)

## Abstract

### Background

Healthcare innovations have not kept pace with the burden of critical illness survivorship. The majority of patients treated in an intensive care unit (ICU) will survive but suffer new or worsening physical, cognitive or mental health sequelae, known as post-intensive care syndrome (PICS). For these survivors, the transition from hospital-based acute care to community-based care is often complex, with high rates of emergency department visits and unplanned hospital readmission. The purpose of this analysis is to describe ICU survivor and family caregiver experiences navigating the challenges in the transition from hospital to home.

### Methods

In this qualitative interpretive description study, data from semi-structured interviews with ICU survivors and family caregivers in the months following discharge from the hospital to home were analyzed using thematic and constant comparative methods.

### Results

The 47 study participants included 28 survivors (mean age 58, 17 men and 11 women) and 19 family caregivers (mean age 53, 6 men and 13 women), who represented 32 cases. The challenges experienced when transitioning from hospital included (1) feeling too ill to go home and pushed out of the hospital without a plan, (2) confronting illness and exhaustion without a safety net, and (3) managing at home with inadequate healthcare. During this time, patients were vulnerable to stagnation

**Data availability statement:** There are ethical restrictions on sharing the de-identified transcripts that were the data for this study. The transcripts in their entirety contain potentially identifying information. Data cannot be made publicly available because public availability would compromise patient privacy. Participants did not consent to their de-identified transcripts being shared. The University of British Columbia Behavioural Research Ethics Board and the Fraser Health Ethics Board approved the study protocol, including the consent procedures. If there is a request for data, please contact the Manager of The University of British Columbia Behavioural Research Ethics Board, Maria Valente, at maria.valente@ubc.ca.

**Funding:** This research was funded by the Canadian Institutes of Health Research (CIHR): https://cihr-irsc.gc.ca/e/193.html AWD-016294. A.F.H. holds a Scholar Award and S.C. a Trainee Award from Michael Smith Health Research British Columbia: https://healthresearchbc.ca/. The funders had no role in study design, data collection and analysis, decision to publish, or preparation of the manuscript.

**Competing interests:** R.C.A. has received Honoraria from Edwards LifeSciences and HLS Therapeutics Inc. R.C.A. is on the Advisory Board for Renibus Therapeutics Inc. All honoraria and board membership are for work unrelated to this manuscript. No other authors have any potential competing interests to disclose. This does not alter our adherence to PLOS ONE policies on sharing data and materials. No other authors have any potential competing interests to disclose.

or deterioration of their mental and physical health, unmet healthcare needs, and unplanned emergency department visits and rehospitalization.

## Conclusions

The challenging transition from the hospital setting suggests a heightened window of vulnerability in the initial months post-discharge and emphasizes a crucial missing middle in our healthcare system, leaving vulnerable patients at risk for ongoing and new health problems.

## Introduction

Approximately 80% of critically ill patients treated in an intensive care unit (ICU) survive to hospital discharge [1,2]. Yet, many suffer persistent, new, or worsening physical, cognitive, psychiatric, or social complications, termed post-intensive care syndrome (PICS) [3–5]. Physical sequelae can include respiratory impairment, neuromuscular complications, and ICU-acquired weakness, frequently encumbering daily activities [6,7]. Cognitive impairment and psychiatric conditions, such as anxiety, depression, and post-traumatic stress disorder are common [8–12], greatly reducing health-related quality of life [4,13]. Family members often become informal caregivers and may suffer health issues themselves, known as PICS-Family [14–16].

Healthcare innovations are essential to address the burden of issues related to critical illness survivorship, as survivors rarely have access to programs targeting PICS or healthcare professionals with related expertise [5,17–19]. As a result, urgent care health service utilization is substantial, frequently culminating in emergency department visits and unplanned hospital readmission. In Canadian studies, 38–65% of ICU survivors were re-admitted to hospital within a year of hospital discharge [1,2,20], aligning with pooled data indicating a 53% (95%CI: 44–62%) readmission rate within the same timeframe [21]. Compared to non-ICU hospital patients, ICU survivors have also been reported to be high users of medical resources, home care support, and mental health services, and require rehabilitation and long-term care placement [1,2,20,22].

Recent research from the Society of Critical Care Medicine's international Thrive program underscores challenges faced by ICU survivors and caregivers post-hospital discharge. Qualitative analyses revealed a spectrum of physical, emotional, cognitive, practical, and socioeconomic issues, compounded by limited awareness of PICS, insufficient communication, and difficulties navigating the health system [23–26]. While primarily conducted in the context of American ICU recovery programs, these challenges persist in the absence of such programs, along with patient dependence, evolving caregiver roles, unsupportive and inappropriate home environments and difficulties recovering at home because of the absence of post-ICU care pathways, fragmented healthcare, and limited rehabilitation access [18,27–30]. The Thrive program insights emphasize the importance of emotional and practical support, aftercare initiatives, and community-based

resources [23,25]. Considering the early hospital discharge phase, Brown et al., [19] suggested focusing on medication management, medical equipment, subspecialty medical care, rehabilitation, promoting healthy habits, and addressing cognitive impairment and mental health issues.

Alongside documented patient and caregiver needs and suggestions for improving recovery, capturing insights from patients and families within the specific healthcare context is essential for informing the development of patient-centred post-ICU interventions, care pathways, and approaches to care [2,31,32]. This research was conducted in the province of British Columbia, Canada, where there is a universal, publicly-funded healthcare system and no formal post-ICU aftercare or recovery programs. Our overarching comprehensive qualitative study aimed primarily to develop explanatory models of post-ICU unplanned rehospitalization in British Columbia as the foundation for healthcare service redesign in our province through interviews with ICU survivors, family caregivers, and healthcare providers. Challenges in the transition from hospital to home featured prominently in the data from this overarching study. Thus, we conducted a secondary analysis aiming to describe ICU survivor and family caregiver experiences of navigating challenges in the transition from hospital to home and the impact on after-discharge hospital utilization. Our focus on the transition from hospital to home is rooted in the Canadian setting of public healthcare services, the absence of post-ICU programs or services in our context, and the needs of local administrators responsible for healthcare service development and delivery.

## Methods

### Design

A qualitative, interpretive description methodology was utilized for the overarching study and this secondary analysis focused on the transition from hospital to home. Interpretive description is a practical approach to gathering the accounts of participants and constructing experiential evidence that is relevant and useful for real-world clinical settings [33]. Rather than rely on a theoretical framework, in keeping with interpretive description methodology, we prioritized findings that are relevant and potentially useful for knowledge users in clinical settings. Aligning with the principles of patient-oriented research and integrated knowledge translation, the research team included an interdisciplinary group of researchers, clinicians, stakeholders/decision-makers, and patient partners (individuals who were a patient or a caregiver with lived experience) as equal team members engaged throughout all research phases [34,35]. In addition to proficiency in the methodology being used, the research team had clinical and research expertise in critical care, post-ICU recovery, illness survivorship, chronic illness, and health services development and evaluation, and represented the professions of nursing, medicine, and genetic counselling. The University of British Columbia Behavioural Research and Fraser Health Ethics Boards approved the study protocol (H21-01378).

### Setting and participants

This research was conducted in the province of British Columbia (BC), Canada. Participants were recruited from a closed 46-bed ICU (where patients are admitted and cared for by intensivists) of a large urban hospital serving a population of 900,000. There were no formal ICU aftercare, follow-up, or recovery programs in the province at the time of the study. Recruitment began on November 1, 2021 and closed on February 20, 2023. The recruitment process involved purposive sampling where point-of-care providers, specifically physicians, nurses, or social workers, approached eligible patients or the eligible patient's substitute decision maker while in the ICU or after transfer to the hospital ward. Purposive sampling involved the intentional inviting of participants who point-of-care providers anticipated would be able to reflect on and articulate their experiences during recovery. The research team maintained regular communication with point-of-care providers throughout the recruitment period to support the inclusion of participants with diverse experiences and variation in sociodemographic and medical characteristics. As recruitment progressed, providers were encouraged to identify and invite potential participants whose perspectives or attributes were underrepresented in the emerging sample, ensuring a

broader range of experiences were captured. If the patient or substitute decision-maker agreed, the provider gave contact information to the research team. The research team then contacted the patient or substitute decision maker by telephone or email within a month of the patient's hospital discharge to discuss their interest in participating and ask whether they would like to be sent a copy of the verbal consent form by email or mail. A research team member obtained informed consent from study participants after reviewing information about the study's aims, risks and benefits, and procedures and answering their questions. Verbal consent was documented by a research team member on the consent form for each participant. An interview was scheduled for a time and date selected by the participant and when they indicated they would have privacy for the conversation. Eligible patients included those who received invasive or non-invasive mechanical ventilation for ≥ 48 hours during their primary ICU admission, were ≥ 19 years of age, spoke English, and could give informed consent at the time of data collection. Individuals receiving palliative care at home or who underwent cardiac surgery were excluded on the assumption that they were receiving related follow-up services. Eligible family caregivers included a relative (an adult child, sibling, spouse, son-in-law, or daughter-in-law), unmarried partner, neighbour, or close friend.

## Data collection

For the overarching study, in-depth semi-structured interviews were conducted with patients and family members from December 2021 to May 2023. Six team members (A.F.H., K.L., J.C., B.H., A.E., and L.B.) conducted the interviews following extensive training by research team members with qualitative research expertise (A.F.H., K.L.). The majority of interviews were conducted virtually (via Zoom Video Communications, Inc.), owing to restrictions on in-person research related to the COVID-19 pandemic. We aimed to interview patients and family members within approximately 1 month of discharge home, either from an initial ICU stay or hospital readmission following a prior ICU admission. A subset of 6 patients who had been readmitted to the hospital were interviewed in-person in their hospital room. One patient participant wrote and emailed his responses to the interview questions. Depending on patient and family preferences, the interviews were conducted either independently or with the patient and family member together as a dyad interview.

Interview guides were developed for the overarching study drawing from our prior research with critical illness survivors, the literature (on recovery following critical illness, post-intensive care, and hospital readmission), and the input of our patient and caregiver partners, and clinician research team members. We revised the interview guides extensively based on revisions suggested by patient and caregiver partners and clinician team members. Directed by the overarching study aims, the interview guides queried patient and family member health-related priorities, the individual and relationship impact of illness, the challenges and stressors encountered since hospital discharge, strategies used to manage, healthcare services or resources sought, and for those readmitted to the hospital, the challenges that contributed to their rehospitalization (see S1 File). In line with interpretive description and qualitative semi-structured interviewing techniques [33], these interview guides were not rigidly followed but instead functioned as a tool to facilitate conversation and encourage participants to share their experiences. Interviewers guided discussions by encouraging elaboration on aspects of illness and recovery raised by participants. Additional questions that arose during discussions were also posed. Adhering to an iterative approach, we adjusted the interview guides based on preliminary insights identified during ongoing analyses. Interviews with individual patients and with patient-caregiver dyads lasted on average 43 minutes (range 17–104 minutes), were audio-recorded, transcribed verbatim, de-identified and cleaned for accuracy. Interviewers made field notes after each interview, documenting overall impressions and highlighting aspects of participants' experiences that were emphasized or unique.

Consistent with interpretive description, we focused on gathering interview data high in information power [36]. This approach offers an alternative perspective on determining the appropriate sample size of a qualitative study, distinct from data saturation. Information power indicates that the more information the sample holds for the study, the fewer participants needed [36]. We deemed the sample and data used for the analysis presented here to be high in information power

because there was wide variation in participants' characteristics (e.g., patient admitting diagnosis, participant sociodemographics, and the nature of the patient-caregiver relationship) and the types of experiences they described, as well as the richness of participants' accounts and the high quality of interview dialogue. Self-reported sociodemographic data were collected via Zoom following the interview or self-report via a Research Electric Data Capture (REDCap) survey link. Medical data were extracted from patient charts.

## Data analysis

Our analysis was guided by the applied analytic direction of interpretive description, involved research team input throughout, and the use of data management software NVivo™ version 12. For the overarching study, an initial coding frame was inductively identified, discussed, and developed based on 10 interviews, at which time it became apparent that the transition from hospital to home was particularly difficult for patients and families. Thus, for the secondary analysis, an inductive in-depth analysis of this broad theme on data from 17 patients and 10 family caregivers was undertaken. We identified transcript segments that reflected emergent patterns and examples, which were then grouped and regrouped into codes and categories, and then themes based on the process of comparing and contrasting data across participants. These codes, categories, and themes were then applied to the remaining patient and caregiver transcripts to further develop and refine the analysis. This data coding was an inductive and iterative process (versus one focused on inter-coder reliability) completed by four team members who regularly met with the research team to discuss the application of the coding frame, suggestions for revisions to the frame, and new insights gleaned. The ongoing comparing and contrasting of participants' experiences and the ideas within the categories was facilitated by intentionally aiming for a higher level of interpretation, that is, moving from descriptive analysis to interpretive analysis [33]. Diagramming of emerging findings via online whiteboard and visual collaboration using Miro™ complemented the coding of data.

The four-member in-depth analysis group, including two trainees who were involved in conducting interviews (B.H., A.E.,) and two healthcare professional team members with extensive qualitative research expertise (A.F.H., K.L.), discussed the ongoing analysis and emergent findings during bi-weekly meetings. The evolving analysis was shared with the entire research team who engaged in open dialogue and discussion that enabled an interpretive analysis informed by the multiple perspectives of our patient partners and multidisciplinary clinical stakeholders. This approach was an alternative to conventional member checking, which is discouraged in interpretive description due to its potential to sway interpretation and hinder the formation of meaningful findings [33]. Further, team meetings functioned as reflexive spaces wherein our assumptions, disciplinary backgrounds, and lived experiences complemented one another in shaping the analysis, while also helping to surface and address potential analytic biases. Through this iterative and dialogic engagement, the team co-constructed findings that were analytically rigorous and grounded in experiential insight.

## Patient partner involvement

Our researcher team included 4 individuals with lived experiences as ICU survivors and 3 as family caregivers of ICU survivors. Their involvement greatly shaped the study conduct and interpretation. Patient partners were engaged throughout all research phases, including through dedicated patient partner meetings, full team meetings, specific data analysis sessions, and manuscript development. We invited individuals to provide input on study aspects that aligned with their interests and welcomed input in their preferred formats. Patient partners reviewed study materials and offered suggestions for revisions, including recruitment protocols and documents and interview guides. One patient partner opted to review and clean interview transcripts and offered analytic insights during data analysis meetings. As data collection progressed, patient partners provided input into the consistency and resonance of the provisional insights and findings with their own experiences, which raised questions for ongoing iterative data collection and analysis and were used to refine the findings. Draft findings were circulated for feedback, and patient partners provided written and verbal input that gave direction

for subsequent analysis and writing. They also reviewed manuscript drafts and offered valuable suggestions that strengthened the final interpretations and presentations of results.

## Rigor

The rigor of this interpretive description study was enhanced by involving patient partners, clinicians, and healthcare stakeholders in all phases. We revised the study materials with feedback from patient partners and clinicians to bolster credibility and ensure we gathered meaningful data. The research team collaborated on analyzing data and critiquing preliminary findings to enhance representative credibility, analytic logic, and interpretive authority. Ongoing dialogue enhanced the team's reflexivity. An audit trail, including meeting minutes, analytic notes and diagrams, and draft analysis materials, documented methodologic decisions and evolving analyses. The study findings are reported following the Consolidated Criteria for Reporting Qualitative Research guidelines and checklist [37].

## Findings

Of the 47 participants, 28 (59.6%) were patients and 19 (40.4%) family caregivers. These participants represented a total of 32 unique patient cases (Fig 1).

Patients were aged 19–85 years (mean 58), with 60.7% identifying as men, 60.7% were of European cultural background, 60.7% living with a support person, and 92.3% living in a city. Most patients were currently not employed (85.7%) and indicated some level of difficulty living on their current household income (64.3%). Family caregivers were aged 24–80 years (mean 53), with 68.4% identifying as women, 47.4% were of European cultural background, 63.2% living with the patient, and 94.7% living in a city. Over half were currently employed (63.2%) and 52.5% indicated some level of difficulty living on their current household income (see Table 1 for participant demographics). Of the 32 patient cases, 59.4% had experienced readmission to the hospital. The median APACHE II score was 19 (IQR 9.25) and ranged from 2 to 19. The most common primary admitting diagnoses were sepsis or septic shock (31.2%), acute respiratory disorders (18.8%), and pneumonia (12.5%), and 21.9% of patients were COVID positive. Over half (59.4%) received invasive mechanical ventilation. Median ICU length of stay was 10 days (IQR 12.5) and ranged from 3 to 10 days, while median hospital length of stay was 42.5 days (IQR 29) and ranged from 3 to 153 days. Median days from ICU discharge to study interview was

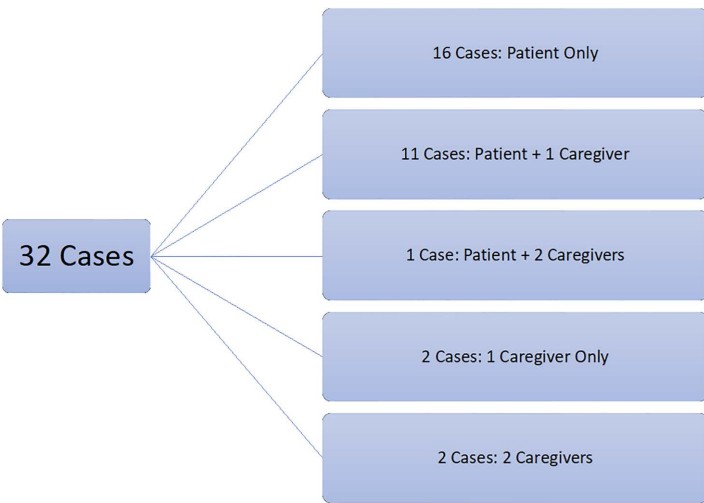

**Fig 1. Breakdown of participant cases.**

**Table 1.  Patient and caregiver participant characteristics.**

| Characteristic | Patient (n = 28) Number (%) | Caregiver (n = 19) Number (%) |
|---|---|---|
| Age mean years (range) | 58 (19–85) | 53 (24–80) |
| Gender | | |
| Woman | 11 (39.3) | 13 (68.4) |
| Man | 17 (60.7) | 6 (31.6) |
| Relationship to Caregiver (Your caregiver is your…) | | |
| Spouse/Common-law Partner | 14 (50) | |
| Parent | 3 (10.7) | |
| Sibling | 4 (14.3) | |
| Adult child | 3 (10.7) | |
| Other | 4 (14.3) | |
| Relationship to Patient (You are the patient's...) | | |
| Spouse/Common-law Partner | | 8 (42.1) |
| Parent | | 3 (15.8) |
| Sibling | | 2 (10.5) |
| Adult Child | | 6 (31.6) |
| Marital Status | | |
| Single | 7 (25) | 4 (14.3) |
| Married/Common-law/Living with Partner | 14 (50) | 14 (73.7) |
| Divorced | 6 (21.4) | 1 (5.3) |
| Widowed | 1 (3.6) | |
| Living Arrangement | | |
| Living alone | 7 (25) | |
| Living with patient/support person | 17 (60.7) | 12 (63.2) |
| Other | 4 (14.3) | 7 (36.8) |
| Cultural Background* | | |
| African | | 1 (5.3) |
| European | 17 (60.7) | 9 (47.4) |
| East Asian | 2 (7.1) | |
| South Asian | 2 (7.1) | 3 (15.8) |
| South East Asian | 1 (3.6) | 3 (15.8) |
| First Nations or Indigenous | 3 (10.7) | 1 (5.3) |
| Other | 3 (10.7) | 3 (15.8) |
| Community Size | | |
| Rural Community with less than 1,000 residents | | 1 (5.3) |
| Small Town with between 1,000 and 29,999 residents | 2 (7.1) | |
| Medium-Size City with 30,000–99,999 residents | 7 (25) | 7 (36.8) |
| Large City with 100,000 or more residents | 19 (67.9) | 11 (57.9) |
| Highest Level of Education Completed | | |
| Some secondary/high school | 8 (28.6) | |
| Completed secondary/high school | 3 (10.7) | 3 (15.8) |
| Some college or university | 8 (28.6) | 2 (10.5) |
| Completed a graduate or professional degree | 8 (28.6) | 13 (68.4) |
| You don't have an option that suits me | 1 (3.6) | 1 (5.3) |
| Employed Before Critical Illness | | |

*(Continued)*

**Table 1.** (Continued)

| | Patient (n = 28) | Caregiver (n = 19) |
|---|---|---|
| Yes | 14 (50) | 15 (78.9) |
| No | 14 (50) | 4 (14.3) |
| Additional Sources of Income Before Critical Illness | | |
| Employment Insurance (EI) | 1 (3.6) | 2 (10.5) |
| Social Assistance | 4 (14.3) | |
| Disability Benefits | 7 (25) | 1 (5.3) |
| Pension (includes any type/source of pension) | 7 (25) | 3 (15.8) |
| Employed now | | |
| Yes | 4 (14.3) | 12 (63.2) |
| No | 24 (85.7) | 7 (36.8) |
| Additional Sources of Income Now | | |
| Employment Insurance | 1 (3.6) | 2 (10.5) |
| Social Assistance | 3 (10.7) | 0 |
| Disability Benefits | 8 (28.6) | 2 (10.5) |
| Pension (includes any type/source of pension) | 10 (35.7) | 5 (26.3) |
| Difficulty Living on Total Household Income Before Critical Illness | | |
| Not at all difficult | 14 (50) | 12 (63.2) |
| Somewhat difficult | 8 (28.6) | 5 (26.3) |
| Difficult | 6 (21.4) | 2 (10.5) |
| Difficulty Living on Total Household Income Now | | |
| Not at all difficult | 10 (35.7) | 8 (42.1) |
| Somewhat difficult | 8 (28.6) | 6 (31.6) |
| Difficult | 10 (35.7) | 4 (21.1) |
| No response provided | | 1 (5.3) |
| Level of Activity Now | | |
| Usual activity – no problem | | 16 (84.2) |
| Mild change – able to continue normal activity | 2 (7.1) | 1 (5.3) |
| Change in usual activity – bed rest less than 50% waking hours | 9 (32.1) | 1 (5.3) |
| In bed/chair more than 50% waking hours | 10 (35.7) | 1 (5.3) |
| Bed/chair ridden or unable to care for self | 7 (25) | |
| Access to Home Care Support Now | | |
| Respite | 4 (14.3) | 1 (5.3) |
| Home care Registered Nurse calls | 2 (7.1) | |
| Home care Registered Nurse visits | 4 (14.3) | 1 (5.3) |
| Support worker visits (e.g., bathing) | 5 (17.9) | 1 (5.3) |
| Pharmacy delivers medications | 3 (10.7) | |
| Pharmacy bubble packs meds | 3 (10.7) | 1 (5.3) |
| Equipment (e.g., wheelchair, walker, bed or mattress) | 13 (46.4) | 3 (15.8) |
| Other | 5 (17.9) | 3 (15.8) |

* One caregiver participant identified 2 cultural background categories.

49 (IQR 77) and ranged from 0 to 350 days. This wide range reflects diversity in time spent in non-ICU hospital wards (see Table 2 for additional patient medical data).

## 1. Feeling pushed out without a plan

The accounts of ICU survivors and families consistently included recollections that, prior to hospital discharge, they thought the patient was too ill to go home, and the feeling that they were being pushed out of the hospital despite unclear plans for managing illness and recovery at home.

### Too ill to go home

Many of the patients and family caregivers described thinking the patient was too ill to go home, questioning whether the patient's *"diagnosis was complete,"* and their medical issues adequately investigated and treated. Several expressed doubt that the patient had recovered from their primary illness sufficiently, naming ongoing severe symptoms and missed opportunities for investigations and treatment of unresolved disease.

**Table 2. Patient medical characteristics.**

| Characteristic | Patient Cases (n=32)<br>Number (%) |
|---|---|
| APACHE II Score, median (IQR) | 19 (9.25) |
| Primary ICU admitting diagnosis | |
| Sepsis or septic shock | 10 (31.2) |
| Acute respiratory disorders | 6 (18.8) |
| Pneumonia | 4 (12.5) |
| Cardiac arrest | 2 (6.3) |
| Gastrointestinal bleed | 2 (6.3) |
| Neurological disorders | 2 (6.3) |
| Other | 6 (18.8) |
| COVID positive | 7 (21.9) |
| Ventilation | |
| Invasive mechanical ventilation | 19 (59.4) |
| Non-invasive mechanical ventilation | 12 (37.5) |
| Other | 1 (3.1) |
| ICU length of stay (days), median (IQR) | 10 (12.5) |
| Hospital length of stay (ICU plus ward) (days), median (IQR) | 42.5 (29) |
| Days from ICU discharge to study interview, median (IQR) | 49 (77) |
| Polypharmacy | |
| Before hospital admission | 19 (59.4) |
| At hospital discharge | 24 (75) |
| Primary care provider identified | 28 (87.5) |
| Re-admission to hospital after ICU stay | 19 (59.4) |

The findings suggest that the transition from hospital to home was particularly challenging, with participant commentaries conveying vulnerability regarding stagnation or deterioration of patients' mental and physical health, unmet healthcare needs (delay or nonreceipt of needed healthcare), and unplanned rehospitalization (Fig 2).

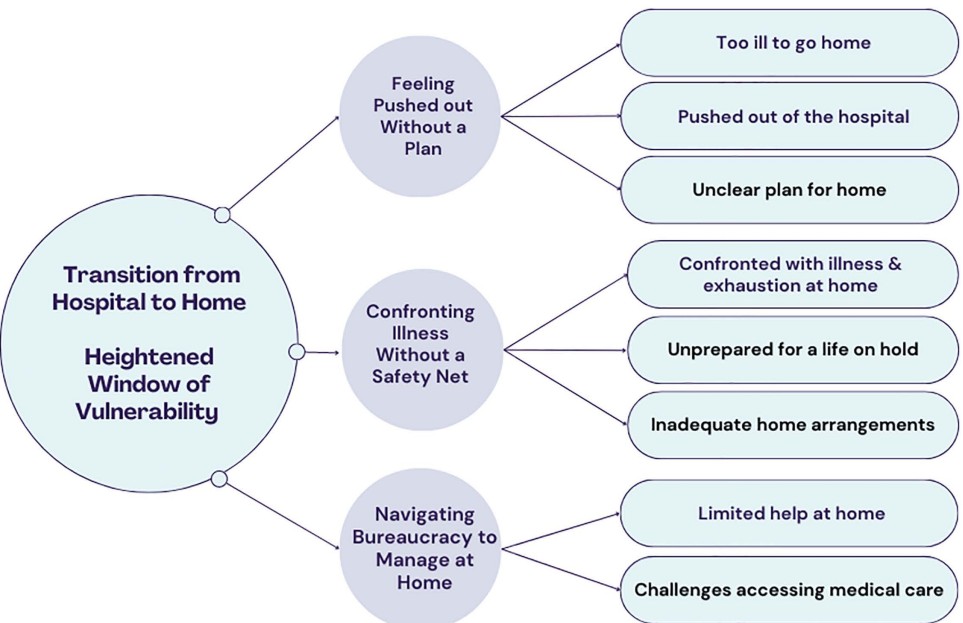

**Fig 2. Overview of challenges in the transition from hospital to home.**

*I think they sent her home too early without fully investigating the problem. The bacteria thing. At that point in time, she might have been able to have a biopsy done.* (79-year-old man caring for his 76-year-old wife admitted to ICU with urosepsis)

Others described the primary medical problem as taking priority but with disregard for additional health problems, especially those for which there was no clear diagnosis, treatment, or management plan.

*Before I left the hospital, my stomach was feeling weird, and I thought it was related to the feeding tube. When I brought it up before I left, it was kind of dismissed. Then a month later I found out it was a stomach ulcer. So that was a little frustrating. And then also there was some weird numbness in my hands and my arms and every time I brought it up before I left the hospital, no one investigated.* (35-year-old man admitted to ICU with urosepsis)

The patients and family caregivers described the patients' bodies as markedly changed, extremely physically weak, and fatigued before discharge. The participants reported significant symptoms such as breathlessness, pain, swelling, limb numbness and nerve damage, unhealed wounds or incisions, and muscle and body aches, collectively experienced as total exhaustion.

*My body. Oh, I had absolutely no strength. I felt like a child. I would get up with my walker to go to the bathroom and it was like I was a child taking my first steps. I was so weak.* (76-year-old woman admitted to ICU with urosepsis)

Apparent in participant commentaries were the patient's mobility and functional struggles in repositioning and getting out of bed, walking independently, or sitting in a chair for any length of time. The patients and family caregivers were aware of the physical toll illness had taken and the patient's lack of energy and strength to tolerate their multiple symptoms without assistance or medical care.

Feeling too ill to go home, many of the participants conveyed a sense of worry, uncertainty, anxiety, and distress with the patient's looming hospital discharge. Neither the patients nor the caregivers could envision how the patient's medical needs would be met, how they would function safely at home, or how the family would manage their care. This was exacerbated in situations where the patient lived alone. In one instance, a patient who had been readmitted to the hospital described the unbearable uncertainty and fear of being home alone immobile, suffering in pain, and unable to carry on with normal daily living activities. Rather than being discharged home again, this factored into this patient's decision to have medical assistance in dying, which is legal in Canada.

## Pushed out of the hospital

Many participants indicated that they thought the patient was pushed out of the hospital before they were well enough, or the home environment was adequately set up. Some recounted dehumanizing commentary by staff that the patient was taking up a bed needed by others and that *"it takes $1,500 to take care of you every day"*, leading them to conclude that hospital staff had come to see them as a statistic and using resources that could be more usefully deployed. Other participants described being made to feel guilty that they were still ill, responsible for not recovering quickly enough, and therefore undeserving of continued hospital care.

Despite receiving the message that discharge was all-important to hospital staff, several participants indicated the inadequacy of discharge preparation, including rehabilitation, providing information or education, and arranging home care or support and medical follow-up. The participants perceived rehabilitation as essential to gaining enough strength and function to be at home and expressed tremendous frustration that this was largely insufficient in the hospital, was left too late and rushed at the end based on a predetermined date rather than when the patient was ready. Some described constant pushing from staff but with little consideration for how the patient was tolerating the discharge preparation. Rather, discharge suitability appeared to be based on healthcare providers' conceptualizations of medical stability and a narrow focus on the patient's ability to perform limited tasks (e.g., walking a certain number of steps) in the hospital, which were not reflective of the degree of functioning required to be safe at home.

*They kept rushing me, "We have to get you out of here because there are other people waiting for these beds, and we have to get you motivated..." Then, they took me once around the block walking. I didn't feel very good. And I said, "I don't feel very good." And they said, all the medication you'll feel better and this, that, and the other. Then I went home and I was very upset and very nervous… And I thought, oh my goodness, what happens in the night if I get ill? And an abundance of worry and having felt I was thrown out.* (85-year-old woman admitted to ICU for COVID-19 pneumonia)

Despite multiple attempts by some family caregivers to advocate for the patient by voicing their opinion that the patient was not well enough to go home, they commonly reported feeling dismissed and disregarded. They also felt that healthcare providers were ignoring family caregiver concerns that they could not simply *"just up and leave my job"* to assume the caregiver role.

*They were ready to discharge him. And I was like, "What are you talking about? He still is not that mobile". And actually, it got quite ugly with them back and forth. I said, "I'm not bringing him home. He's in no condition to be home." I did everything I could. I spoke up… In the end, they just said, "He's coming home." They arranged a [public disability transport], and he got dropped off at home… It was on Tuesday, I was at work."* (44-year-old-woman caring for her 80-year-old father admitted to ICU with hospital-acquired pneumonia)

Advocating for delayed discharge or specific supports to be arranged before being sent home was a common frustration reported by these family caregivers. Some described themselves as being like a broken record, repeating their

concerns so many times to so many different healthcare providers, few of whom listened. Others indicated they did not know what to say or whom to talk to about their concerns about the patient's readiness to go home or their own ability to provide the requisite support and care.

In contrast to the family caregivers who felt pushed out of the hospital, there were ICU survivors who themselves pushed for discharge. They described weighing staying in the hospital for medical reasons and the fear that something could go wrong at home against the comforts of home as a place for rest and recovery. Others appeared to be motivated primarily by their desire to attend important events, though in retrospect, they sometimes expressed regret. For example, a 73-year-old woman who had survived pneumonia and sepsis indicated she had *"pushed too hard"* and had to be readmitted to the hospital *"because it didn't turn out like I'd hoped, and I wasn't ready to be out."*

### Unclear plan for home

Many of the patients and caregivers remarked on how little information and education they received while the patient was in the hospital and how unclear communication had been about medical follow-up and support available once home. Not knowing what to expect left many feeling unprepared to leave the hospital, feelings that were worsened by the mental toll of having survived a life-threatening illness, worry about the fragility of their health, and being *"terrified something's going to happen again"* at home. The participants wished they had received information and education on symptoms to watch for, appropriate action to manage symptoms, an anticipated timeframe for recovery, and whom to contact with questions or concerns.

> *It would've been nice [to have been told] what to expect from recovering with liver and kidney issues – these are the symptoms you might experience. It wasn't really outlined… I didn't know what to expect.* (35-year-old man admitted to ICU with urosepsis)

Many were discharged home with outstanding questions about wound and incision care, medications, rehabilitation exercises, and appropriate ways to function safely at home. Feelings of being unprepared, worried, and fearful became magnified once the patient was home.

> *One thing that's been a challenge is all her stitches and stuff because there's a lot of care involved in taking care of those, and they didn't really give us any instructions. So, we've just been winging it, and there's obviously a lot of stress about it getting infected. We don't know if it looks right or it doesn't…. We haven't had any sources given to us yet.* (39-year-old woman caring for 19-year-old daughter admitted to ICU with traumatic chest injury)

The patients and caregivers also commonly described their perception that staff expected family members to take on responsibility for arranging and providing any and all types of care at home.

### 2. Confronting illness without a safety net

While the participants' narratives suggested that the patient was confronted with their illness and exhaustion at home, the home environment was inadequate for recovery and left many patients feeling unprepared for life on hold.

### Confronted with illness and exhaustion at home

When faced with having to function with the reality of a changed body in their home environment, participants described how the countless physical and emotional challenges contributed to a crushing sense of weakness, exhaustion, and inability to function and manage independently. Many feared they would not make it more than a day or two before returning to the hospital, and at least 7 patients had a fall resulting in rehospitalization. While some ICU survivors and family

caregivers emphasized ongoing symptoms, others explained how their ill health limited their ability to function day-to-day in their homes. *"Overwhelmed to see the degree of loss he had in the hospital,"* they commonly struggled to prepare or eat meals and use the washroom for toileting or bathing.

*The day I got here [home] I remember I honestly felt I was going back to the hospital. I was so weak. My body didn't move, I had a hard time even grabbing a glass of water.* (52-year-old man admitted to ICU with sepsis)

Mobilizing in the home and leaving the house for shopping, exercise, or maintaining quality of life was daunting and seemed out of reach for many.

*I can cook and do things, but the longer I stand on my legs, the longer the pain goes in, it's in my toes. And then it goes up above my knees… nerves maybe. You're trying to maybe stay off your feet, sitting down. None of it seems to work…. It's a life changer when your legs don't work.* (48-year-old man admitted to ICU with empyema)

The participants also noted that the extent of the patient's deterioration had been somewhat obscured in the hospital, where they had constant monitoring and care, compared to at home, where they had to be relatively self-sufficient. As such, many were confronted with anxiety, worry, and fear with day-to-day living, particularly when venturing outside the house.

*I've been trying to go to the mall, and I'll get very overwhelmed very easily… I get very bad anxiety…. Obviously, I'm nervous of hurting myself or something going wrong. But then I've noticed a lot of like social anxieties… where like in crowds I'll feel suffocated and I think that goes back to the fear of not being able to breathe. And then I get scared of not having access to the medication or [a place to] rest as soon as I start feeling that pain.* (19-year-old woman admitted to ICU with traumatic chest injury)

For many of these patients and family caregivers, feeling stuck in a downward spiral of poor physical functioning, low energy, inability to carry out daily activities, anxiety and fear, and loss of interest in meaningful activities led to a feeling of complete physical and mental exhaustion.

**Inadequate home arrangements**

Because the patients and family caregivers had been unaware of what life would be like for the patient at home, arranging support or making home modifications before discharge was impossible. Thus, many portrayed their home as being inadequate for daily living, recovering, or healing. This was particularly true for patients who lived alone, without family members or close friends to help.

*When you get home from the hospital, all the food in your fridge is no good. So now you got to go and do some shopping and that's where it draws you down. Because you're not ready for this anxiety… What food do I need? It's all rotten; got to throw all this out…. You got nobody taking care of you, but you come out of there, you're [going to have to be] self-sufficient.* (69-year-old man admitted to ICU with respiratory failure secondary to COVID-19 pneumonia)

Momentous efforts were required to meet basic needs, often demanding families reconfigure their homes and juggle caregiving duties with their other responsibilities. Many participants devised solutions or workarounds to meet the patient's immediate basic needs, some of which became immensely difficult for both the patient and the caregiver.

*I'm not able to eat on my own, which has been quite challenging, especially since my sister works. So, we've been trying to find ways for me to get around that, like drinking smoothies or shakes during the day… She [caregiver] works*

*full work hours... So usually, she helps me go to the washroom before she leaves, and I don't drink anything until she's going to be home. Otherwise, I'll have to go to the washroom again. Oh my, I've been stuck.* (35-year-old man admitted to ICU with urosepsis)

Characterizing their situation as an impossible one, some family caregivers highlighted instances when they were forced to make distressing choices between caring for the patient or attending to other responsibilities, creating risks that could jeopardize the patient's health.

*I left him [patient] here [home] for about two hours… I was in an impossible box where I couldn't take him with me... The only way I could get through it was to accept the fact that it was possible that I would come home, find him fallen down the stairs.* (44-year-old-woman caring for her 80-year-old father admitted to ICU with hospital-acquired pneumonia)

They feared the potential consequences for the patient, including a fall, inadequate hydration and nutrition, further physical deconditioning, social isolation and loneliness, depression, and even a life-threatening event. Therefore, instead of being a time of recovery, for many, the initial transition to home was recalled as a time of dwindling health, further physical and emotional decline, and a window of extreme vulnerability.

## Unprepared for life on hold

In the weeks following hospital discharge, the patients and family caregivers continued to be taken aback by how far below baseline the patient's physical health had fallen, their inability to return to their prior health, and the slow progress in their recovery. Few had known what recovery might look like or benefitted from a comprehensive plan.

*I didn't know if the way I was feeling was normal. The exhaustion and the physical weakness… I just didn't know a lot about the recovery process and I didn't have anybody to really ask.* (59-year-old woman admitted to ICU with COVID-19 pneumonia)

In retrospect, both the patients and caregivers commonly conveyed that they had held high expectations that they, or their loved one, would immediately *"bounce back"* to how their life was before hospitalization. Yet, the reality of recovery in most instances was felt to be much longer and not to the extent anticipated, if at all. Given the ongoing physical burden of illness and slow recovery, several patients and family caregivers described a marked emotional toll that included frustration, fear, uncertainty, and sadness.

Continuing dependence on family caregivers was necessary for many patients, who expressed frustration and a sense of helplessness with their loss of independence and ability to function as previously.

*It was [challenging for her] going up the stairs. It got a bit more normal doing that. But for her it was probably a little bout of depression… She was upset that she wasn't able to do things. And she's not motivated to do things, so not getting better.* (30-year-old man caring for wife admitted to ICU with pneumonia and sepsis)

Several patients expressed worry and guilt about *"burdening"* and *"putting too much pressure"* on their loved ones, given the extent and ongoing nature of the assistance they required. For ICU survivors, who had been previously employed, the protracted recovery forced a change in employment due to ongoing physical limitations and health issues. The sudden and unanticipated need to re-evaluate their career and the possibility of never being employable again further contributed to feeling frustrated, worried, and emotionally drained.

*Now I'm going to have to completely depend on my family for everything. I can't work or anything. So now I have to ask like a child again.* (39-year-old woman caring for 19-year-old daughter admitted to ICU with traumatic chest injury)

Moreover, ongoing health issues and slow recovery left some patients and caregivers worried that the patient would miss out on significant life events, such as family life cycle celebrations and seasonal gatherings. In all, marked physical and functional impairment, unresolved health issues, slow recovery, ongoing need for care, uncertainties around employment and participating in meaningful family life, and the high emotional burden, left many patients feeling that, *"my life is basically on hold right now."*

### 3. Managing at home with inadequate healthcare

Once home, the participants were faced with managing their health but in the context of inadequate healthcare related to limited help at home and challenges accessing medical care.

### Limited help at home

The patients and caregivers recalled unclear or vague plans for medical follow-up, support, or resources made before discharge home. While many indicated that referrals had been made to home care, physiotherapy, occupational therapy, or to arrange equipment (e.g., oxygen, walker, wheelchair, special mattress), there were numerous accounts where participants *"couldn't get a hold of anyone,"* referral plans did not materialize, or the initial contact, home visit, or appointment was significantly delayed, or rather difficult. Moreover, in-home assessments for necessary supports and services were described as confusing, where the participants highlighted difficulties qualifying for and then obtaining the required help. Many were baffled by systems and processes that made no sense, seemed ineffective, and left them with little understanding of how to proceed.

*The people from home care came. The girl was asking me for a copy of my 2021 tax assessment and I didn't have it… And she says, well, can you give me three months of bank statements? I went to my computer and I printed out a whole bloody year… Then they contacted me and said the bank statements didn't do, they need an assessment... And then the whole issue kind of died because [they] never got back to us.* (79-year-old man caring for his 76-year-old wife admitted to ICU with urosepsis)

In several instances, there also appeared to be a mismatch between the supports on offer, primarily limited home care, and those that the patient or family deemed necessary.

*Every question I've asked of the case manager, it's been, "Well, I don't know. I don't know how that works." And it's been very, very, to be honest, frustrating.* (44-year-old woman caring for her 67-year-old father admitted to ICU with cardiac arrest)

For some, this lack of clear and timely follow-up and the appropriate type of assistance left patients and caregivers feeling frustrated and abandoned at home, with nowhere to turn when needing help. There was a sense of resigned acceptance of inadequate help.

*He's been back to his GP and it's just like, "Don't worry, it's just [hand] numbness. It'll disappear." But it doesn't. It's getting worse… It's like you start having these things where, yeah, we'll send someone, we'll cover for this. And, he [patient name], he's pretty sharp, he can follow up on stuff. But he's not going to initiate, he's not going to call back to the hospital and say, "Hey, I heard someone is going to follow up on these three things and they haven't yet. So who do I talk to?" That's not going to happen.* (69-year-old woman caring for 59-year-old husband admitted to ICU with seizures)

## Challenges accessing medical care

Unclear expectations for recovery, coupled with the lack of a follow-up plan, left most patients and family caregivers guessing at what care they needed to arrange and feeling frustrated, anxious, and distressed. Given the high level of exhaustion and confusion, a few participants felt unable to figure out the next steps of their recovery. Others used trial and error in managing the ongoing health problems because they did not have ready access to healthcare services and were left to problem-solve on their own.

*The blood pressure's high, all of a sudden it's gone high and you think, oh no, have I done something wrong? And then you try to figure that out, what do I do now? … We've bought three different blood pressure machines thinking something's wrong with one... You don't want to have what happened before, so you're making sure you're checking everything off. Lately, I get a little bit worried when something is not quite right, sort of antsy.* (61-year-old man caring for 61-year-old wife admitted to the ICU with pneumonia and sepsis)

When health issues remained unresolved or new problems arose, the patients who had access to a primary care provider or allied healthcare provider (physiotherapist, occupational therapist, counsellor) were grateful. However, even those with a healthcare provider encountered difficulties scheduling a timely appointment and figuring out who the right primary or specialist provider was, particularly considering the patient's complex health issues and in the context of uncoordinated and siloed services.

*I was discharged from the hospital without a follow-up of any sort from anyone about my feeding tube. I called my doctor. He said, "Well, call the guy who did the surgery." I called the hospital, left messages with the record department... I contacted four different agencies during that time… nobody could tell me who was going to remove the stitches around my belly for the feeding tube. That went on for two or three weeks. I did not get an answer from anywhere.* (58-year-old man admitted to the ICU with community-acquired pneumonia)

Additional challenges in accessing care initially following hospital discharge included the momentous effort of arranging transportation and travelling when the patient was exhausted and coping with severe physical impairments.

*Getting me to like doctor's appointments or even getting me to the hospital was really difficult to the point where we had to have someone who could carry into the building… So, a lot of times I would like just not go to appointments or we'd have to reschedule. So, a lot of those kind of things got delayed… Just getting me places has been a lot harder… Having to schedule them and like get transportation was probably the hardest thing.* (35-year-old man admitted to ICU with urosepsis)

Those without an established primary healthcare provider were often at a total loss for how to access care. Some tried to use their personal connections, some asked healthcare providers to make exceptions, and others tried to work the system any way they knew how, often to no avail.

## Discussion

Uniquely embedded in a publicly-funded universal healthcare system, our findings point to poor hospital discharge preparation, inadequate home environments and community services, and heavy reliance on family caregivers. During this time, patients were particularly vulnerable to stagnation or deterioration of their mental and physical health, delay or nonreceipt of needed healthcare (i.e., home care support, rehabilitation, primary care provider appointments), and unplanned emergency department visits and rehospitalization. These findings suggest a heightened window of vulnerability in the initial weeks and months after discharge – characterizing this as a crucial time that sets the stage for recovery.

Meyer et al., [38] noted that a continuous arc of recovery following a critical illness is reflected poorly in the design of modern healthcare systems, and fragmented care is epitomized at the crucial time of discharge home. Our study revealed not only fragmented care in our Canadian context, but a marked discrepancy between the needs of survivors and the available healthcare services, where many perceived their discharge to be premature and the requisite preparation wholly insufficient. Healthcare providers had deemed these patients not well enough to be hospitalized, but the patients commonly felt too unwell to be home. This finding surfaces the mismatch between patient and provider perspectives regarding the severity of illness post-ICU requiring medical care and hospitalization – a novel finding with this population. We also found that accessing medical care and rehabilitation in the community was challenging for most. These results echo a 2018 Scottish study, which has a public healthcare system, that identified poor preparation specific to critical illness survivors for hospital discharge, poor communication and coordination between acute and community care, and lack of timely support upon returning [32]. However, our findings extend existing literature focused on a lack of coordination and timely access, emphasizing difficulties facing patients as well as their family caregivers when they do not have a primary care provider and community support and rehabilitation appear nonexistent.

Our findings are likely reflective of the markedly strained Canadian healthcare system at the time we conducted the study, when already stark health human resource shortages in acute and primary care and a growing population requiring care became even more pronounced during the COVID-19 pandemic and have largely persisted. That is, broader systemic healthcare system challenges existed before the pandemic but these were also exacerbated during the pandemic. Before the pandemic and even now, people are discharged from the hospital as soon as they are deemed medically ready to go home, and they are not given a choice to extend their stay. We postulate that during the course of the study, hospital clinicians had limited time and resources to prepare for the transition home and that caseloads for home care nurses and other support services prevented effective coordination or timely care. These findings were evident in interviews conducted during and at the tail end of the pandemic. Thus, the fragility of the existing system was fully exposed, given what we know about the complex ongoing health issues, high risk for PICS, and increased health service use for this population.

Of concern in our analysis were the multiple reports of the basic and functional needs of the survivor not being met once home, even with family member assistance, because of challenges in performing activities of daily living (ADL) (i.e., toileting, bathing, feeding) as well as the more complex instrumental activities of daily living (IADL) (i.e., meal preparation, house cleaning, driving, shopping, and medication management). Several participants linked their ADL, IADL, and functional disability directly to health decline or incidents that landed them back in the hospital. These findings add in-depth descriptions that greatly complement the mounting evidence that has identified impaired ADL, IADL, and functional outcomes [39–48]. A 2017 review by Hopkins et al. [45] reported that across 4 studies, 69% of people treated in an ICU had new or worsening IADLs that persisted months to years following ICU discharge. ADL and functional disability have also been associated with an increased risk of rehospitalization, institutionalization, mortality, and higher healthcare spending [22,44,48]. This raises concern for survivors in general, but even more so for those who do not have a family caregiver to help at home and/or who are socioeconomically disadvantaged and lack access to material resources. Recent international studies have identified critical illness as a modifier of social determinants of health [24,26], and our study uniquely illustrates how employment and financial strain among patients and caregivers shape recovery even in a country with publicly-funded health and social services. These findings are novel and important considering that caregiver strain can exacerbate symptoms of PICS-Family [15].

In our research, poor discharge and transition home and the inability to access help once home were clearly traumatic for several survivors, thus extending existing evidence of the many sources of fear, anxiety, and distress following critical illness. Also prominent in our study was the described caregiver emotional strain associated with their attempts

to assist survivors, even when many caregivers had minimal preparation and competing responsibilities. These experiences seem to represent mechanisms by which survivor and family mental health worsens, which are novel findings. While emotional support for patients and family caregivers has been identified as key to enhancing critical illness recovery [19,23,24], post-ICU healthcare services that meet patient and caregiver needs might go a long way to improving mental health.

## Implications

It might be possible to remedy emotional, physical, functional, and cognitive PICS sequelae by integrating transitional care interventions, ICU aftercare clinics, and home-based programs into Canadian healthcare services. Transitional care interventions span from hospital to home in the form of care pathways, home visit programs, and structured telephone support [49,50]. Outpatient ICU aftercare, follow-up, or recovery clinics vary in their management (e.g., led by nurses, physicians, or a team), clinic focus (e.g., physical and/or mental health), consultation or counselling format (e.g., direct contact or phone), frequency and dose of follow-up (e.g., monthly or weekly), and eligible patients (e.g., duration of ICU stay or specific diagnosis). Recommendations for home-based exercise and physiotherapy-led interventions have also been developed [51]. While transitional care interventions [52–54], ICU aftercare clinics [48–51], and home-based programs [54–58] have yet to demonstrate effectiveness in trials, this is likely reflective of the heterogeneity of critical illness survivors, the need for integrated interdisciplinary care, and the earlier stages of intervention refinement [58–60]. For example, qualitative evidence from the Society of Critical Care Medicine's Thrive initiative, including ICU aftercare and peer support, demonstrated marked effects on patients' mental and physical health [23,61,62]. McPeake et al., [62] suggested that successful ICU recovery programs had five key components: continuity of care, improving symptom status, normalization and expectation management, internal and external validation of progress, and reducing feelings of guilt and helplessness.

Tailoring and integrating transitional care interventions, ICU aftercare clinics, and home-based programs to our local context and our patient population needs, in step with robust process evaluations, are necessary to improve post-ICU recovery. Members of the Thrive Post-ICU Collaboratives also recommend the assessment of patient pre-ICU functional abilities at ICU admission and hospital discharge, as well as serial assessments of PICS-related problems within 2–4 weeks of hospital discharge as the basis for referrals to services [63]. Efforts to map and integrate these recommendations into existing and new healthcare services are warranted. Doing so will require innovative solutions that recognize the finite resources available in the fiscally strained healthcare climate, the health human resource shortage, and the large number of Canadians living in rural and remote locations or unable to secure transportation. For example, Rosa et al. [64] argued that in-person clinic-based aftercare requiring survivors to arrange and take transportation might deprive the most disabled individuals and those lacking pragmatic social support, thereby contributing to health inequities. Thus, a flexible complex intervention approach, incorporating elements of transitional care, aftercare clinics, and home-based programs, seems most likely key to meeting the diversity and needs of a complex and heterogeneous population.

## Strengths and limitations

Participants were recruited from a single, albeit large, tertiary care hospital; we do not claim our results are generalizable. Instead, as is characteristic of qualitative evidence, these findings provide insights that are transferrable to other post-ICU contexts or settings. Only individuals who were approached by point-of-care providers were invited, which could have influenced the type of participants recruited. Our study participants may have been slightly sicker than those typically included in post-ICU literature, given the 10-day median ICU length of stay and 43-day median hospital length of stay. This may have contributed to the extent and severity of physical and functional challenges patients encountered once home and the high degree of caregiving deemed necessary. Study recruitment occurred during the COVID-19 pandemic, severely limiting our team's capacity to enrol participants and significantly extending recruitment timelines. Many survivors

and family members also declined study participation once they were home, indicating they were too exhausted and over-whelmed to participate in an interview. Thus, our sample likely does not represent the full range of survivor or family caregiver experiences. It is also possible that the extra COVID-19-related demands on the healthcare system impacted these study participants' experiences and our findings. However, we found that the accounts of participants interviewed throughout the course of the study were similar, including when the healthcare system was no longer as taxed from the pandemic. Moreover, findings highlighting the insufficiency of healthcare services and the need for comprehensive post-hospital follow-up echo other pre-pandemic Canadian studies [65,66]. Our reliance on virtual interviews limited study participants to those with a phone or computer and who could navigate scheduling. We were struck by the high number of participants for whom phone participation appeared difficult, who did not have voicemail, whose voicemail was full, or who struggled to remember the scheduled interview.

Despite these limitations, our study was bolstered by the robust variation in survivor demographics and ICU admitting diagnoses, family caregiver demographics and relation to the survivor, as well as experiences described by participants. Rigor was further enhanced by the richness of information garnered through in-depth interviews with both survivors and family caregivers [36] and interpretation that incorporated clinical expertise [33]. The inclusion of patient partners as research team members is increasingly recognized as crucial for producing evidence relevant to patients' priorities. This research was deeply reflective of the contributions of our clinician and survivors and family caregiver partners who enhanced the team's reflexivity and the representative credibility, analytic logic, and interpretive authority of the findings.

## Conclusion

Despite the significant progress in critical care for treating illnesses, the nature of the challenges experienced by participants in this research suggests a need for healthcare services that facilitate patient recovery after hospitalization, specifically for ICU survivors. The challenging transition from the hospital setting emphasizes a crucial missing middle in the Canadian healthcare system, leaving vulnerable patients at risk for ongoing and new health problems.

## Supporting information

**S1 File. Interview guides.**
(DOCX)

## Acknowledgments

We would like to thank Myles Lynch, Bernard Lynch, Cameron Rankin, our other patient and caregiver partners on our Critical Illness Survival Patient Advisory Board, Hiroki Sato, our other multidisciplinary research team members, and our participants for their support on this project.

## Author contributions

**Conceptualization:** Amanda Fuchsia Howard, Gregory Haljan.

**Formal analysis:** Amanda Fuchsia Howard, Kelsey Lynch, Sally Thorne, Leanne M. Currie, Rakesh C. Arora, Robert C. McDermid, Omar Ahmad, Sarah Crowe, Sybil Hoiss, Anita David, Joanne Chu, Alice Erchov, Brianna Hou, Arnima Singh, Miki Tsui, Gregory Haljan.

**Funding acquisition:** Amanda Fuchsia Howard, Gregory Haljan.

**Methodology:** Amanda Fuchsia Howard, Sally Thorne.

**Project administration:** Kelsey Lynch.

**Supervision:** Amanda Fuchsia Howard.

**Writing – original draft:** Amanda Fuchsia Howard, Gregory Haljan.

**Writing – review & editing:** Amanda Fuchsia Howard, Kelsey Lynch, Sally Thorne, Leanne M. Currie, Rakesh C. Arora, Sarah Crowe, Sybil Hoiss, Alice Erchov, Brianna Hou, Arnima Singh.

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
