## [Decision Letter · Decision Letter 0]

23 May 2025

PONE-D-24-49915From hospital to home: A heightened window of vulnerability post-critical illnessPLOS ONE

Dear Dr. Howard,

Thank you for submitting your manuscript to PLOS ONE. After careful consideration, we feel that it has merit but does not fully meet PLOS ONE’s publication criteria as it currently stands. Therefore, we invite you to submit a revised version of the manuscript that addresses the points raised during the review process.

Please ensure you respond to and address the queries, changes, comments and suggestions offered by the reviewers. 

We look forward to receiving your revised manuscript.

Kind regards,

Allen Prabhaker Ugargol

Academic Editor

PLOS ONE

Journal Requirements:

“This research was funded by the Canadian Institutes of Health Research (CIHR): https://cihr-irsc.gc.ca/e/193.html AWD-016294. A.F.H. holds a Scholar Award and S.C. a Trainee Award from Michael Smith Health Research British Columbia: https://healthresearchbc.ca/.”

“R.C.A. has received Honoraria from Edwards LifeSciences and HLS Therapeutics Inc. R.C.A. is on the Advisory Board for Renibus Therapeutics Inc. All honoraria and board membership are for work unrelated to this manuscript. No other authors have any potential competing interests to disclose.”

**Additional Editor Comments:**

Dear Authors,

I have received review feedback from both reviewers. Based on the reviewers comments and my own assessment, I am offering you the decision of 'minor revision' for this manuscript and encourage you to comply with the queries, comments and suggestions of the reviewers. Please find attached the reviewers comments and suggestions.

Reviewers' comments:

Reviewer's Responses to Questions

**Comments to the Author**

1. Is the manuscript technically sound, and do the data support the conclusions?

Reviewer #1: Yes

Reviewer #2: Yes

2. Has the statistical analysis been performed appropriately and rigorously? 

Reviewer #1: N/A

Reviewer #2: N/A

3. Have the authors made all data underlying the findings in their manuscript fully available?

Reviewer #1: No

Reviewer #2: No

4. Is the manuscript presented in an intelligible fashion and written in standard English?

Reviewer #1: Yes

Reviewer #2: Yes

5. Review Comments to the Author

Reviewer #1: The manuscript presents a qualitative interpretive descriptive study focusing on the challenges faced by patients and family caregivers during the transition from hospital to home following critical care. By examining the physical, emotional, and logistical difficulties encountered during this critical recovery period, the study addresses an important gap in post-ICU care and offers insights that could inform the development of more effective models of transitional care and support systems for survivors and their families.

Below are specific points to consider for enhancing the manuscript:

1) In order to further emphasize the originality of the work, it would be beneficial if the extent to which the results go beyond the existing literature were made clearer (e.g., inclusion of patient and provider perspectives)

2) It could be helpful to explicitly describe how potential biases (e.g., those arising from participant selection or the influence of team discussions) were minimized to further strengthen the interpretative authority of the findings. For instance, how were participants selected? Was the selection process random? How was it ensured that only "dissatisfied" individuals did not disproportionately participate in the interviews? Alternatively, this could be acknowledged as a limitation of the study.

3) However, it would be helpful to explore in greater detail how the pandemic may have influenced the participants' experiences. Were the challenges described exacerbated by the pandemic, or do they reflect broader systemic issues?

Reviewer #2: This is a qualitative study exploring key challenges and issues for patients and carers of critical care survivors, focussing on the transition of care between hospital and home.

The methodology is robust and described in detail. The authors used a pragmatic qualitative approach to analysis, which was embedded in the real word experience of people. The methods are described in detail. The analytic method was iterative and driven by the desire to understand and describe the issues from the patient/relative perspective.

The presentation of results is detailed but well presented. Th derivation of the key themes and sub-themes are well-supported by quotes. The authors note they considered reflexivity, and also used early cases t adapt information gathering and enquiry.

I enjoyed reading this article. It is to some extent context specific, which is reflected on by the authors in terms of grounding in the BC healthcare system. The findings are somewhat shocking, but not discordant with other literature.

The results are summarised nicely in the graphic. The discussion is detailed and appropriate in terms of strengths and weaknesses, and provides detailed discussion in context of existing literature. The implications generally, and especially locally, are well-presented.

I only have a few suggestions and queries

1. The authors mention purposive sampling of ICU survivors, but not what they meant by this. Did they employ a framework against patient characteristics etc? More detail would be useful.

2. I was unclear how this analysis of their data was framed in a larger study? Can more information about this be presented. It was not clear if this was the primary purpose of the study, or whether this is a secondary question analysed from the interviews.

3. It would be interesting and useful to note the background and experience of the four person analyst group. Were they clinicians, for example? How did this relate to the reflexivity and exploration of potential analyst bias etc?

4. The authors note that patients were involved in the study, including in the eview and development of the emerging themes. However, it is not clear how they were involved at different stages. Can more information be provided, for example in a specific section about patient involvement?

6. PLOS authors have the option to publish the peer review history of their article (what does this mean? ). If published, this will include your full peer review and any attached files.

**Do you want your identity to be public for this peer review?** For information about this choice, including consent withdrawal, please see our Privacy Policy .

Reviewer #1: No

Reviewer #2: **Yes: ** Timothy S Walsh

---

## [Author Response · Author response to Decision Letter 1]

17 Jun 2025

Response to Reviewers

We thank the editor and reviewers for their thoughtful comments and suggestions for revising our manuscript. We have revised our manuscript accordingly, with our responses and revisions described below.

Reviewer #1:

The manuscript presents a qualitative interpretive descriptive study focusing on the challenges faced by patients and family caregivers during the transition from hospital to home following critical care. By examining the physical, emotional, and logistical difficulties encountered during this critical recovery period, the study addresses an important gap in post-ICU care and offers insights that could inform the development of more effective models of transitional care and support systems for survivors and their families.

Below are specific points to consider for enhancing the manuscript:

1) In order to further emphasize the originality of the work, it would be beneficial if the extent to which the results go beyond the existing literature were made clearer (e.g., inclusion of patient and provider perspectives)

• Response: Throughout the discussion we added language to make it more explicit where our findings were novel or unique when compared to existing literature.

• In the strengths/limitations section (page 38) we also added 2 sentences relevant to the inclusion of patient, family caregiver, and clinicians as research team members.

2) It could be helpful to explicitly describe how potential biases (e.g., those arising from participant selection or the influence of team discussions) were minimized to further strengthen the interpretative authority of the findings. For instance, how were participants selected? Was the selection process random? How was it ensured that only "dissatisfied" individuals did not disproportionately participate in the interviews? Alternatively, this could be acknowledged as a limitation of the study.

• Response: In the sampling and recruitment section (page 7), we added detail specific to the purposive approach we used.

• Given the nature of purposive recruitment and sampling, we also added the limitation this may have created for the study (page 37).

3) However, it would be helpful to explore in greater detail how the pandemic may have influenced the participants' experiences. Were the challenges described exacerbated by the pandemic, or do they reflect broader systemic issues?

• Response: We bolstered the 3rd paragraph of the discussion (page 33, 34) to address these comments more extensively.

Reviewer #2

This is a qualitative study exploring key challenges and issues for patients and carers of critical care survivors, focussing on the transition of care between hospital and home.

The methodology is robust and described in detail. The authors used a pragmatic qualitative approach to analysis, which was embedded in the real word experience of people. The methods are described in detail. The analytic method was iterative and driven by the desire to understand and describe the issues from the patient/relative perspective.

The presentation of results is detailed but well presented. Th derivation of the key themes and sub-themes are well-supported by quotes. The authors note they considered reflexivity, and also used early cases t adapt information gathering and enquiry.

I enjoyed reading this article. It is to some extent context specific, which is reflected on by the authors in terms of grounding in the BC healthcare system. The findings are somewhat shocking, but not discordant with other literature.

The results are summarised nicely in the graphic. The discussion is detailed and appropriate in terms of strengths and weaknesses, and provides detailed discussion in context of existing literature. The implications generally, and especially locally, are well-presented.

I only have a few suggestions and queries

1. The authors mention purposive sampling of ICU survivors, but not what they meant by this. Did they employ a framework against patient characteristics etc? More detail would be useful.

• Response: In the sampling and recruitment section (page 7), we added detail specific to the purposive approach we used.

• Given the nature of purposive recruitment and sampling, we also added the limitation this may have created for the study (page 37).

2. I was unclear how this analysis of their data was framed in a larger study? Can more information about this be presented. It was not clear if this was the primary purpose of the study, or whether this is a secondary question analysed from the interviews.

• Response: We added clarifying language in several places, including the last paragraph of the introduction (page 5, 6), and the beginning of the data collection (page 8) and analysis page 10, 11) sections.

3. It would be interesting and useful to note the background and experience of the four person analyst group. Were they clinicians, for example? How did this relate to the reflexivity and exploration of potential analyst bias etc?

• Response: In the data analysis section (page 11, 12) we added details of the smaller analysis group and described interactions with the larger team as it related to reflexivity and potential bias.

4. The authors note that patients were involved in the study, including in the review and development of the emerging themes. However, it is not clear how they were involved at different stages. Can more information be provided, for example in a specific section about patient involvement?

• Response: We added a stand-alone paragraph/section to the methods (page12) describing patient partner involvement throughout the study.

---

## [Decision Letter · Decision Letter 1]

23 Sep 2025

From hospital to home: A heightened window of vulnerability post-critical illness

PONE-D-24-49915R1

Dear Dr. Howard,

We’re pleased to inform you that your manuscript has been judged scientifically suitable for publication and will be formally accepted for publication once it meets all outstanding technical requirements.

Kind regards,

Taiwo Opeyemi Aremu, MD, MPH, PhD

Academic Editor

PLOS ONE

Additional Editor Comments (optional):

Reviewer #2:

Reviewers' comments:

Reviewer's Responses to Questions

**Comments to the Author**

1. If the authors have adequately addressed your comments raised in a previous round of review and you feel that this manuscript is now acceptable for publication, you may indicate that here to bypass the “Comments to the Author” section, enter your conflict of interest statement in the “Confidential to Editor” section, and submit your "Accept" recommendation.

Reviewer #2: All comments have been addressed

2. Is the manuscript technically sound, and do the data support the conclusions?

Reviewer #2: Yes

3. Has the statistical analysis been performed appropriately and rigorously? 

Reviewer #2: N/A

4. Have the authors made all data underlying the findings in their manuscript fully available?

Reviewer #2: Yes

5. Is the manuscript presented in an intelligible fashion and written in standard English?

Reviewer #2: Yes

6. Review Comments to the Author

Reviewer #2: (No Response)

7. PLOS authors have the option to publish the peer review history of their article (what does this mean? ). If published, this will include your full peer review and any attached files.

**Do you want your identity to be public for this peer review?** For information about this choice, including consent withdrawal, please see our Privacy Policy .

Reviewer #2: **Yes: ** Professor Tim Walsh, Usher Institute, University of Edinburgh

---

## [Editor Report · Acceptance letter]

PONE-D-24-49915R1

PLOS ONE

Dear Dr. Howard,

I'm pleased to inform you that your manuscript has been deemed suitable for publication in PLOS ONE. Congratulations! Your manuscript is now being handed over to our production team.

Kind regards,

on behalf of

Dr. Taiwo Opeyemi Aremu

Academic Editor

PLOS ONE